# Oxidative Stress and Inflammation in Osteoporosis: Molecular Mechanisms Involved and the Relationship with microRNAs

**DOI:** 10.3390/ijms24043772

**Published:** 2023-02-14

**Authors:** Teresa Iantomasi, Cecilia Romagnoli, Gaia Palmini, Simone Donati, Irene Falsetti, Francesca Miglietta, Cinzia Aurilia, Francesca Marini, Francesca Giusti, Maria Luisa Brandi

**Affiliations:** 1Department of Experimental and Clinical Biomedical Sciences “Mario Serio”, University of Florence, Viale Pieraccini 6, 50139 Florence, Italy; 2F.I.R.M.O. Italian Foundation for Research on Bone Disease, Via San Gallo 123, 50129 Florence, Italy

**Keywords:** oxidative stress, osteoporosis, inflammatory mediators, miRNAs, bone remodeling

## Abstract

Osteoporosis is characterized by the alteration of bone homeostasis due to an imbalance between osteoclastic bone resorption and osteoblastic bone formation. Estrogen deficiency causes bone loss and postmenopausal osteoporosis, the pathogenesis of which also involves oxidative stress, inflammatory processes, and the dysregulation of the expression of microRNAs (miRNAs) that control gene expression at post-transcriptional levels. Oxidative stress, due to an increase in reactive oxygen species (ROS), proinflammatory mediators and altered levels of miRNAs enhance osteoclastogenesis and reduce osteoblastogenesis through mechanisms involving the activation of MAPK and transcription factors. The present review summarizes the principal molecular mechanisms involved in the role of ROS and proinflammatory cytokines on osteoporosis. Moreover, it highlights the interplay among altered miRNA levels, oxidative stress, and an inflammatory state. In fact, ROS, by activating the transcriptional factors, can affect miRNA expression, and miRNAs can regulate ROS production and inflammatory processes. Therefore, the present review should help in identifying targets for the development of new therapeutic approaches to osteoporotic treatment and improve the quality of life of patients.

## 1. Introduction

Bone remodeling is due to the highly regulated and coupled events of bone resorption and bone formation in which old bone is removed and replaced with newly formed bone. It occurs before birth and throughout life, and its role is to maintain bone strength and mineral homeostasis [1]. Bone is a dynamic tissue, and bone remodeling is crucial not only to replace primary with secondary bones, but also to repair skeleton microdamage and ensure normal calcium homeostasis. Osteoclasts and osteoblasts are involved in bone remodeling, and have the sequential function of resorbing old bone and forming new bone through a well-coordinated mechanism regulated by different factors, such as systemic hormones, local growth factors, cytokines, chemokines, adhesion molecules, and extracellular matrix proteins [1]. These factors are secreted by bone surface cells and produced by osteocytes located in the bone matrix, which represent the other cell type that, together with bone lining cells, is involved in bone remodeling [2].

In particular, osteoblasts arise from bone-marrow-derived stromal cells (BMSCs), pluripotent mesenchymal stem cells, with the ability to differentiate into chondrocytes and adipocytes. The differentiation of BMSCs towards osteogenesis is regulated by several bone morphogenetic proteins (BMPs) and Wnt signaling pathways [3] through the activation of β-catenin, which is stabilized by the parathyroid hormone involved in calcium homeostasis [4]. Active osteoblasts synthetize and secrete Type I collagen (COLI), the principal bone matrix protein and noncollagenous proteins, such as osteocalcin (OCN), osteopontin, and osteonectin, and are rich in alkaline phosphatase (ALP), a biomarker of osteoblastic function that is very important in the mineralization of bone matrix [5,6]. In addition, osteoblasts control osteoclastogenesis through both the Wnt signaling pathway, and the secretion of osteoprotegerin (OPG) and the receptor activator of nuclear kB ligand (RANKL). In fact, OPG downregulates osteoclastogenesis, while RANKL upregulates it [7].

Mature osteoclasts are multinucleate cells that derive from the monocyte/macrophage lineage and through the fusion of mononuclear precursor cells. They adhere to the bone matrix, polarize, and present on distinct plasma-membrane domains with different functions. In particular, the ruffled border domain, located on the apical membranes, represents the resorbing organ that allows for osteoclasts to degrade hydroxyapatite, the inorganic component of the bone matrix, through the secretion of hydrochloric acid and proteases [8]. RANKL binds to its receptor activator of nuclear kB (RANK), located on the surface of osteoclastic precursors, and induces their differentiation. This does not occur when RANKL binds to its soluble receptor OPG. For this reason, the ratio of OPG/RANKL is an index of the degree of differentiation of osteoclasts and is essential for the dynamic balance of bones in the body [8].

Osteocytes derive from mature matrix-producing osteoblasts, are embedded in bone, play a chief role in maintaining the correct biomechanical property of bone tissue, and present dendritic processes through which they communicate with adjacent osteocytes and other cells on the bone surface. Some important signals that regulate osteoblastic and osteoclastic activity start from osteocytes. In fact, they secrete sclerostin to inhibit Wnt signaling pathway, involved in bone formation, and RANKL, which induces osteoclastic differentiation, in osteoblasts. In addition, osteocytes produce OPG that, by competing with RANKL for binding to the RANK receptor, suppresses osteoclastic activity. The expression of genes related to mineralization and phosphate metabolism also occurs in osteocytes that produce fibroblast growth factor 23 (FGF23), a phosphaturic hormone involved in the regulation of phosphate homeostasis and bone mineralization [9]. In the presence of microdamage, osteocytes undergo apoptosis, which triggers signals that promote the local recruitment of osteoclasts and their activity in repairing bone damage. This is explained by considering that apoptotic osteocytes do not produce OPG, but can stimulate neighboring viable osteocytes to produce RANKL [10]. Moreover, apoptotic osteocytes produce high-mobility group box protein 1, which can enhance the RANKL/OPG ratio, and increase the production of interleukin (IL)-6 and tumor necrosis factor-alpha (TNFα) in BMSCs [11]. However, osteocytes can also undergo autophagy, which, under stress conditions, allows for the elimination of unnecessary organelles for a return to normal cellular function [12].

When the functions of osteoclasts and osteoblasts are unbalanced, the bone-remodeling process may be highly altered, resulting in disorders related to bone metabolisms. In particular, osteoporosis is characterized by increased bone resorption due to excessive osteoclastogenesis and osteoclastic activity that is not balanced by osteoblast-mediated bone formation. This leads to increased bone fragility and fracture risk that can have a significant impact on quality of life [13]. The excessive apoptosis that osteocytes undergo in some pathological conditions is also related to the loss of bone mineral density (BMD) and bone mass, causing osteoporosis [14]. Estrogens are important regulators of bone metabolism and exert a bone-protective role by reducing bone resorption and maintaining bone formation [15]. Therefore, estrogen deficiency in postmenopausal women induces osteoporosis associated with an excessive increment of bone resorption because of an increase in the number and activity of osteoclasts and osteocyte apoptosis [15,16,17]. However, in addition to hormonal changes, other factors are involved in the pathogenesis of osteoporosis, including aging, oxidative stress, and inflammation [18,19,20]. There is increasing evidence that oxidative stress increases with age and/or menopause, and that the lack of estrogen is often accompanied by an increase in reactive oxygen species (ROS) and proinflammatory cytokines such as IL-1, IL-6, and TNFα that, in turn, stimulate ROS production [19,21,22]. A close correlation exists between inflammation and ROS, which are produced as proinflammatory mediators during inflammatory processes. All this can contribute to the activity of bone cells and thereby bone homeostasis by increasing and inhibiting osteoclastic and osteoblastic function [23]. Indeed, both oxidative stress and inflammation can be involved in the development of osteoporosis by preventing the differentiation of osteoblasts, inducing the differentiation and activity of osteoclasts, enhancing apoptotic osteocytes, and increasing the expression of RANKL and the RANKL/OPG-ratio [24,25,26,27,28].

MicroRNAs (miRNAs) are small endogenous single-stranded noncoding RNA molecules containing approximately 22 nucleotides. They recognize specific mRNA sequences and induce the mRNA degradation or block gene expression at post-transcriptional levels by binding to the 3′-untreated region [29]. Since different genes can be targets of one miRNA, and multiple miRNAs can target the same genes, miRNAs regulate biological processes at the network level [30]. miRNAs regulate several physiological processes, including the bone metabolism, by controlling the proliferation and differentiation of osteoblasts and osteoclasts [31,32]. In addition, miRNAs modulate the expression of very important genes involved in the recruitment of various bone cells in the osteogenic process, and facilitate mineralization by affecting the maturation of the collagen fibrillar matrix [33,34]. However, the dysregulation of miRNA expression also plays an important role in the development of disease [35], and can be involved in bone-related diseases and disorders such as osteoporosis [32,36,37]. ROS can down- or upregulate the expression of miRNAs and vice versa, and miRNAs can affect oxidative stress by inhibiting or stimulating ROS production [38]. miRNAs are capable of regulating the function of immune cells and act as promoters or suppressors of inflammatory response [39]. In this review, the principal mechanisms through which oxidative stress and inflammation affect bone mass, and the relationship among inflammation, oxidative stress, and miRNAs in the pathogenesis of osteoporosis are reported. Therefore, the aim of the present review is to identify targets that facilitate the development of new potential therapeutic treatments for osteoporosis.

## 2. Oxidative Stress and Osteoporosis: Principal Involved Molecular Mechanisms 

During normal metabolism, NADPH oxidase localized on the membrane and mitochondrial oxidases are the main endogenous systems involved in the production of superoxide anion (O2^●−^) and hydrogen peroxide (H_2_O_2_). This production of ROS under physiological conditions is usually counterbalanced by adequate antioxidant systems, including vitamins E and C, glutathione peroxidase, reduced glutathione, superoxide dismutase, and catalase [40,41]. When the intracellular production of ROS is controlled, they act as second messengers by regulating and activating signaling transduction pathways involved in numerous biological processes such as apoptosis, survival, differentiation, proliferation, and inflammation [42,43]. Bone repair and bone remodeling are also redox-regulated processes, and the physiological redox state is essential for the equilibrium between osteoblastogenesis and osteoclastogenesis [24,44]. The transient production of ROS due to RANKL into osteoclastic precursor is very important for its role in the induction of osteoclastogenesis, and this indicates that ROS act as intracellular mediators for osteoclastic differentiation [45]. Therefore, while a physiological production of ROS in osteoclasts may be important in facilitating bone destruction and remodeling, osteoblasts produce antioxidant systems to remove ROS released into the surrounding environment [46]. This equilibrium can be altered by oxidative stress, a condition that occurs when high levels of ROS resulting from certain oxidation pathways overcome antioxidant systems. This alteration causes a loss of bone mass and thus osteoporosis [47]. In particular, the excessive production of ROS increases osteoclastogenesis, and reduces osteoblastogenesis and osteoblastic activity, resulting in the altered bone architecture and bone loss that characterize osteoporosis [48].

In primary rat osteoblasts, oxidative stress decreases the nuclear translocation of the transcriptional factor, nuclear factor erythroid-2 related factor (Nrf2), involved in the regulation of cellular response to oxidative stress and the maintenance of bone homeostasis [49]. Nrf2 downregulates the expression of cytokines involved in osteoclastogenesis and induces the transcriptional activation of antioxidant genes; for this, the reduced expression of Nrf2, present in oxidative stress conditions and detected in osteoporotic rats, causes an increase in osteoclastic formation [50,51,52,53]. H_2_O_2_ stimulates osteoclastogenesis by increasing the expression of osteoclastic differentiation factors by activating the transcriptional factor nuclear factor kappa B (NF-kB) and reducing the expression of Nrf2 in mouse mononuclear macrophage (RAW264.7) cells [54,55]. ROS-activated NF-κB can stimulate the expression of transcriptional factors c-Fos and nuclear factor of activated T-cell cytoplasmic 1 (NFATc1), which regulates the genes involved in osteoclastogenesis and bone resorption, such as tartrate-resistant acid phosphatase and cathepsin K. Moreover, ROS induce the expression of an activator of NF-kB, TNF receptor associated factor 6 (TRAF-6), which RANKL recruits through RANK with a consequent increase in osteoclastic formation [55,56]. However, oxidative stress conditions also increase RANKL-induced osteoclastogenesis through the activation of mitogen-activated protein kinases (MAPKs) and NF-kB in osteoclastic lineage cells, leading to an enhancement of bone resorption [57,58].

The activation of extracellular signal-regulated kinases (ERKs) mediates ROS-induced RANKL expression in human osteoblast-like cells [27,59]; RANKL, in turn, increases oxidative stress by activating TRAF-6, Rac1, and NADPH oxidase in the osteoclastic precursor [60]. Taken together, these effects exacerbate osteoclastogenesis and increase bone mass loss.

ROS-activated MAPKs also affect osteogenic differentiation; in fact, the activation of ERKs and c-Jun N-terminal kinases (JNK) due to H_2_O_2_ inhibits osteoblastic differentiation by reducing the expression of osteogenic differentiation markers, such as ALP, OCN, COLI, and Runt-related transcription factor 2 (Runx2) in rabbit calvarial osteoblast, bone marrow stromal cells, and murine preosteoblastic MC3T3-E1 cells [61,62]. Glutathione peroxidase 7 (GPX7), an antioxidant enzyme belonging to the GPX family and located in the endoplasmic reticulum (ER), is involved in the osteogenic differentiation by regulating ER stress and the mammalian target of rapamycin (mTOR), a serine/threonine kinase that controls cellular processes involved in skeletal development and homeostasis. In fact, these signaling pathways can in part mediate the effect of oxidative stress due to the lack of GPX7 in reducing osteogenic differentiation and increasing adipogenesis in hBMSCs and M2-10B4 cells [63,64]. Moreover, oxidative stress can negatively affect osteoblastogenesis through the downregulation of both the Wnt/β-catenin signaling pathway and β-catenin expression [65,66]. In pre-osteoblasts, high ROS levels activate FOXO transcription factors, belonging to the family of forkhead proteins, which, through binding with β-catenin, enhance the transcription of antioxidant enzymes. This apparently positive effect inhibits osteoblastogenesis because it reduces the availability of β-catenin necessary to promote osteoblastic differentiation [67]. Moreover, the activation of FOXO transcription factors due to oxidative stress enhances the expression and the activity of peroxisome proliferator-activated receptor (PPAR)γ, which has the function of stimulating adipogenesis and inhibiting osteogenesis [68]. The transcription factor Krüppel-like factor 5 (KLF5), present in osteoblasts but not in osteoclasts, plays a positive role in the osteogenic differentiation of murine BMSCs through β-catenin. However, the oxidative stress-induced hypermethylation of KLF5, together with its downregulated expression, reduces the expression and nuclear translocation of β-catenin, altering osteogenic differentiation [69]. Moreover, oxidative stress reduces osteoblastogenesis by downregulating the expression and inducing the degradation of the tumor protein p53-inducible nuclear protein 2, which controls the osteogenic differentiation of human BMSCs through the Wnt/β-catenin signaling pathway [70]. Figure 1 summarizes the principal molecular factors involved in the role of oxidative stress on osteoclastic and osteoblastic differentiation.

Excessive ROS production not only induces the defective formation of osteoblasts, but also negatively affects their activity, viability, proliferation, and apoptosis. This leads to a reduction in osteoblastic number and functionality with consequent beginning and evolution of osteoporotic processes [27,71,72,73,74]. The increase in the intracellular activity of ALP and RANKL, and the decrease in IL-6 production due to oxidative stress alter the mineralization processes in H_2_O_2_-treated MC3T3-E1 cells [62,75,76]. Moreover, oxidative stress causes mitochondrial membrane depolarization, ATP level reduction, and apoptosis through JNK activation in osteoblasts [62]. Ca^2+^ influx and Ca^2+^/calmodulin activation due to oxidative stress stimulates the activity of NFATc1 with consequent increase in cell apoptosis and inhibition of mineralization in MC3T3-E1 cells and femoral tissue of osteoporotic female rats [76]. The tripartite motif-containing 33 (TRIM33), belonging to E3 ubiquitin ligases, plays an important role in the differentiation and proliferation of osteoblasts acting as a positive regulator of the BMP-pathway [77]. The expression of TRIM33 is positively related to BMD; in fact, a low expression of this nuclear factor is present in primary human osteoblasts of osteoporotic patients, while TRIM33 overexpression decreases the oxidative stress-induced apoptosis in osteoblasts by inhibiting FOXO3a degradation [78]. Oxidative stress-activated mTOR signaling pathways can also be involved in osteoporosis by increasing the apoptosis of osteoblasts [79]. In fact, blocking the Akt/mTOR signaling pathway inhibits the oxidative stress-induced apoptosis in osteoblasts [73], and the inhibition of mTOR/NF-kB phosphorylation reduces oxidative stress and apoptosis in osteoblasts treated with high levels of glucose [79]. Hyperglycemia is able to reduce bone quality, and diabetic bone disease is a complication of diabetes mellitus in humans [80,81]. In fact, in glucose-treated MC3T3-E1, there is an increase in ROS and apoptosis, along with an upregulation of the expression of activin receptor-like kinase 7. This protein is overexpressed in diabetic rats and appears to be involved in osteoblastic damage due to high glucose levels [82]. The bone loss and osteoporosis induced by glucocorticoids (GCs) are also due to oxidative stress, which negatively affects the proliferation, differentiation, maturation, and apoptosis of osteoblasts, and increases osteoclastic activity [83,84]. GC-induced apoptosis due to oxidative stress is mediated by ERK1/2 activation in MC3T3-E1 cells [85] and involves the downregulation of Nrf2 in BMSCs [86]. It is possible that, in osteoblasts, there is a correlation between the reduced activation of Nrf2 due to oxidative stress [49] and ROS-induced apoptosis, considering that the activation of the γNrf2 signaling cascade inhibits oxidative injury and apoptosis in these cells [87,88].

Given that osteocytes take part in the regulation of bone resorption and formation, the excessive programmed cell death of these cells causes their decrease and represents a crucial aspect in the development of osteoporosis [89]. Postmenopausal osteoporosis is also related to enhanced osteocyte senescence and apoptosis [90,91,92,93]. Indeed, the activation of MAPK signaling due to oxidative stress mediates the increase in apoptosis, RANKL/OPG ratio, sclerostin, and FGF23 levels, and the decrease in autophagy in osteocytic MLO-Y4 cells [28,94,95]. In these cells, the high levels of glucose increase ROS production and apoptosis through the downregulation of the AMP-activated protein kinase (AMPK)/FOXO3a signaling pathway [96]. GCs increase osteocyte apoptosis via the upregulation of NADPH-oxidase, subsequently inducing oxidative stress, as demonstrated in the osteocytes of patients affected by steroid-induced avascular necrosis of the femoral head [73] and in MLO-Y4 cells [97].

Cellular senescence, characterized by the permanent arrest of the cell cycle, participates in age-related osteoporosis, and oxidative stress is one of the factors involved in senescence and senescence-associated secretory phenotype in primary hBMSCs, osteoblasts, and osteocytes [89,91,98,99,100,101].

Given that oxidative stress plays an important role in the pathogenesis of osteoporosis, treatment with antioxidants can improve bone metabolism processes. Many molecules with antioxidant properties, such as flavonoid polyphenols (e.g., genistein, quercetin, and icariin) and nonflavonoid polyphenols (e.g., resveratrol and curcumin), could represent a potential and effective therapeutic treatment of osteoporosis [102,103].

## 3. Proinflammatory Mediators and Osteoporosis: Role of Immune Cells

Inflammation is a defensive response against exogenous and/or endogenous signals. Estrogen loss causes a chronic inflammatory state and increases the levels of proinflammatory mediators, such as cytokines and chemokines, which affect bone cell function, contributing to the development of osteoporosis [18,104]. Indeed, estrogens regulate the expression of cytokines involved in bone cell biology and, in estrogen deficiency, the increase in the production of cytokine levels, such as ILs and TNFα, IL-1β, and interferon-gamma is due to both circulating peripheral blood immune cells and those located in bone tissue [105]. In addition, the altered immune cell number in postmenopausal women and ovariectomized (OVX) rats plays a role in the pathogenesis of osteoporosis [106]. Macrophages, the principal cells involved in the production of cytokines in bone metabolism, are flexible cells with the ability to change function according to their environment, and can reversibly assume the M1 (inflammatory) and M2 (reparative) phenotypes [106,107]. An increase in the M1/M2 macrophage ratio is present in the bone marrow of OVX osteoporotic mice, and estrogen deficiency contributes to the alteration of this ratio by inducing the differentiation of RANKL-stimulated M2 macrophages into osteoclasts. Thus, the M1/M2 ratio may be involved in bone mass loss and considered a potential therapeutic target for the treatment of osteoporosis. [108]. Monocytes, like macrophages, are characterized by different phenotypes and functions, and they remain viable and spontaneously differentiate into osteoclasts in women with postmenopausal osteoporosis in whom there is a higher production of TNFα and RANKL [109]. Peripheral blood mononuclear cells (PBMCs) isolated from female osteoporotic patients undergo spontaneous osteoclastogenesis without exogenous stimulating factors in vitro [110]. Indeed, macrophages and monocytes stimulate the osteoclastic differentiation of PBMCs by producing IL-1 and TNFα [111], and this may explain how inflammation can increase osteoclastic numbers, bone resorption, and osteoporotic processes. The absence of estrogen in OVX mice induces the production of high levels of IL-17 and IL-15 by dendritic cells, which then play a role in inflammation-mediated osteoclastogenesis and bone loss [112]. Moreover, both RANKL and macrophage colony-stimulating factor (M-CSF), involved in osteoclastic differentiation, can stimulate dendritic cells to trans-differentiate into osteoclasts [113], and this can also occur after the direct interaction of dendritic cells with T helper cells (Th) [114]. Since estrogens regulate the function and number of neutrophils, which in turn secrete proinflammatory mediators, these cells can play a role in the development of postmenopausal osteoporosis [106]. The enhancement of RANKL and RANK expression that occurs in inflammatory neutrophils is related to a decrease in BMD and an increase in osteoclastic bone resorption [115,116]. In addition, eosinophils and mastocytes, by producing inflammatory factors, can be associated to the pathogenesis of bone disorders and osteoporosis [117,118]. In particular, IL-31 produced by eosinophils induces osteoclastogenesis through the activation of transcription factors and cytokines correlated to osteoporosis in a condition of inflammatory state, and its levels increase in the serum of postmenopausal women [117]. In the absence of estrogen, mastocytes promote osteoclastic formation by producing TNFα, IL-6, and other mediators with osteoclastogenic properties, and an increase in the mastocyte and osteoclastic numbers occurs in OVX-rodents. Mastocytes may also play an osteoprotective role by producing transforming growth factor-β and IL-12, which stimulate osteoblasts and reduce osteoclastogenesis [119].

The evidence that there is an increase in proinflammatory cytokines involved in the pathogenesis of osteoporosis in estrogen deprivation is demonstrated by the fact that postmenopausal women and OVX rats present high levels of inflammatory cytokines [120,121], significantly reduced by estrogen administration [122,123]. ILs and factors belonging to the TNFα family upregulate the RANK/RANKL pathway in postmenopausal women [124]; in fact, TNFα and IL-1β activate NF-kB and MAPK, enhance the production of RANKL and M-CSF, and downregulate the expression of OPG, resulting in defective bone remodeling [106,125,126,127]. IL-1β increases the expression of RANKL not only in osteoblasts but also in lymphocytes and macrophages, while IL-18 stimulates Th17 cells to release IL-17, which in turn upregulates RANKL [128,129]. However, IL-17 downregulates the expression of Dickkopf-1 (DKK1), an inhibitor of the Wnt signaling pathway and osteoporosis biomarker, in osteoblasts, resulting in an increase in osteoblastic differentiation. Therefore, IL17 may play a dual role, dependent on the presence of other cytokines and the skeletal site [130]. IL-1β inhibits the BMP/SMAD signaling pathway and Runx2 activation, preventing osteogenesis and osteoblastic differentiation, through the activation of NF-kB and MAPKs [131,132]. Although osteoclastic precursors express the IL-6 receptor, IL-6 indirectly induces osteoclastogenesis by promoting RANKL in the osteoblastic precursor and exerts anti-osteoblastic action by inhibiting the Wnt signaling pathway [133,134]. In addition, TNFα inhibits the Wnt pathway through the upregulation of DKK1 expression [135].

Estrogen deficiency activates the nucleotide-binding oligomerization domainlike receptor family pyrin domain-containing 3 (NLRP3) inflammasome, a multiproteic complex involved in immune innate response and inflammation [136]. NLRP3 inflammasome is expressed in osteoblasts and, when it is abnormally activated, plays an important role in the development of osteoporosis. It is activated by the NF-kB-dependent pathway and, after recruitment and activation of caspase-,1, it induces the conversion of pro-IL-1β and pro-IL18 into their mature forms [137,138,139]. The consequence of this is evidenced in the upregulation of osteoclastic differentiation and bone resorption [140,141]. NFLRP3 inflammasome is involved in pyroptosis, a caspase-1-dependent form of programmed lytic cells [142]; in human osteoblast-like cell line MG-63, ROS induce the expression of inflammasome components and promote pyroptosis, resulting in an altered osteoblastic function [143]. The immune system is influenced by the gut microbiota (GM), which in turn affects bone homeostasis through immune and endocrine functions and the maintenance of calcium and phosphate homeostasis. This suggests a possible link among immunity, GM, and bone homeostasis. GM composition changes in the intestine of osteoporotic patients and it may be related to osteoporotic risk. In fact, GM regulates estrogens, and dysbiosis reduces the levels of estrogens into their active form by negatively affecting bone quality and health [144,145]. Probiotic supplementation can promote bone health, and natural anti-inflammatory molecules may be useful in preventing bone loss, the triggering and/or progression of osteoporosis, and fracture risk [146].

## 4. miRNAs and Osteoporosis

miRNAs play a pivotal role during skeletal development, and maintain bone homeostasis by modulating osteoblast–osteoclast axis activity. In fact, miRNAs contribute to the bone metabolism by regulating the differentiation, proliferation, apoptosis, and autophagy of osteoblasts, osteoclasts, and osteocytes [31,98,147]. One major role assigned to miRNAs is to promote the differentiation of BMSCs through the suppression of osteogenic inhibitors or by activating the signaling pathway involved [147,148]. miRNAs, like exosomal miRNAs (Exo-miRNAs), can be found in bioactive vesicles called exosomes, and can be directly secreted by cells as circulating miRNAs (ci-miRNAs) into the blood and biological fluids. Exo-miRNAs can be delivered to the target cells with which they interact to regulate their function, and ci-miRNAs behave like signaling molecules involved in cell-to-cell communication [149,150]. miRNAs also function as mediators between bone cells; in fact, miRNAs released by osteoclasts can affect osteoblastic activity, and those released by osteocytes can regulate the function of other bone cells [151].

Some miRNAs, such as miRNA-33-5p and miRNA-433-3p, promote osteoblastic differentiation via the downregulation of osteoblastogenesis inhibitors [152,153], while others, such as miRNA-194 and miRNA-2861, inhibit osteoblastogenesis through Runx activation [154,155]. In addition, there are miRNAs such as miRNA-34a that inhibit osteoclastogenesis by blocking the pro-osteoclastogenesis factor, transforming growth factor-beta-induced factor 2 [156], or miRNAs such as miRNA-124-3p and miRNA-125a-5p that suppress NFATc1 expression [157,158]. However, several miRNAs, such as miRNA-125b, miRNA-375, and miRNA-133a-5p, can inhibit osteoblastogenic differentiation by acting directly or indirectly on Runx2 [159,160,161]. On the other hand, other miRNAs, such as miRNA-21-5p, miRNA-29, and miRNA-183-5p, can promote osteoclastogenesis by acting on signaling pathways involved in osteoclastic formation and bone resorption [162,163,164]. For this, the dysregulation of miRNA expression is reflected in changes in osteoblastic and osteoclastic differentiation, resulting in altered bone remodeling and the development of bone disease, including osteoporosis [36]. The overexpression of some miRNAs prevents osteoblastic formation and osteogenic differentiation in in vitro cell cultures and in mouse mesenchymal cells, respectively. This occurs through the downregulation of the expression of Runx2 and one of its targets, transcriptional factor Osterix, which completes osteoblastic differentiation and induces the production of bone structural proteins [165,166]. Instead, the reduced expression of miRNAs targeting RANK causes increased bone resorption and reduced bone mass, as observed in OVX rats [167]. Given that higher extracellular miRNA levels are detected in serum from osteoporotic patients compared with that of healthy controls, miRNAs can be considered potential biomarkers of osteoporosis [168].

## 5. Crosstalk among Oxidative Stress, Inflammation, and miRNAs in Osteoporosis

A certain number of miRNAs are able to regulate inflammation, in particular, molecules belonging to pathways involving NF-kB and NLRP3 inflammasome [169]. However, as soon as proinflammatory signals begin, there is a large increase in miRNA expression through the activation of NF-kB [170]. Oxidative stress also plays a critical role in the biogenesis of miRNAs, and ROS can be involved in the deregulation of miRNA expression [38]. With aging, the BMSCs are no longer able to effectively initiate the osteogenic pathway and undergo senescence characterized by inflammation and oxidative stress that can subsequently alter miRNA levels [171]. A relationship among inflammation, ROS, and miRNAs is, therefore, evident in osteoporotic processes. High levels of TNFα due to estrogen deficiency participate in osteoporosis through the downregulation of expression of FOXO1, a very important protein involved in the defense of bones from oxidative damage. In particular, the activation of NF-kB by TNFα promotes the expression of miRNA-705, a post-transcriptional downregulator of FOXO1, with a consequent increase in oxidative damage in osteoporotic mouse BMSCs [172]. TNFα also enhances miR-182, which, in turn, upregulates TNFα-induced murine osteoclastogenesis by inhibiting FOXO3 and the negative regulators of inflammatory osteoclastogenesis, mastermind-like protein 1 [173].

The overexpression of miRNA-128, targeting Sirtuin1 (Sirt1), an inhibitor of NF-kB, is related to an increased inflammatory state and bone resorption in postmenopausal osteoporosis. In fact, in mouse bone-marrow-derived macrophages, high levels of miRNA-128 increase osteoclastogenesis due to low Sirt1 levels and the subsequent activation of NF-kB [174]. Moreover, miRNA-128-3p mediates and exacerbates the inflammatory effect of TNFα, which induces the production of other cytokines by reducing Sirt-1 levels in hBMSCs [175].

During inflammation- and oxidative-stress-induced senescence, an increase in miRNA-141 levels occurs in human BMSCs [176]. The effect of ROS on the upregulation of miRNA-141 was demonstrated in endothelial cells in which the treatment of H_2_O_2_ increases the levels of this miRNA [177]. Moreover, miRNA-141 is related to inflammation via the downregulation of transforming growth factor-beta 2, which behaves as both immunosuppressor and proinflammatory mediator [178]. Zinc metalloproteinase ZMPSTE24 is downregulated when levels of miRNA-141 are high, and this leads to an alteration in bone formation and osteoporosis. In fact, in ZMPSTE24 ^−/−^ knockout mice, good fracture healing cannot occur, and ALP levels decrease [176,179].

miRNA-320a is also overexpressed in osteoporosis, and plays a role on osteoclastogenesis by inhibiting the expression of phosphatase and tensin homolog (PTEN), an inhibitor of AKT expression and the PI3-K pathway [180,181]. The increase in miRNA-320a expression may cause oxidative damage by inhibiting the PI3K/AKT signaling pathway in osteoblasts [182], and it regulates several osteoblastic genes, including those involved in oxidative stress in human primary osteoblasts. For this reason, high levels of miRNA-320a cause alterations in osteoblastic differentiation by lowering mineralization and ALP activity, and increasing ROS production [183].

High levels of ROS present in ferroptosis, a new iron-dependent form of programmed cell death due to lipid peroxidation, alter the function and activity of osteoblasts, osteocytes, and osteoclasts, resulting in loss of bone mass [184] In particular, iron accumulation increases NADPH oxidase 4, an enzyme that enhances ROS levels and intracellular lipid peroxides driving ferroptosis in osteoblasts [185]. The regulation of ferroptosis can prevent osteoporosis, and miRNAs may be a target for the therapeutic treatment of this disease. In fact, several miRNAs whose levels are directly or indirectly related to ferroptosis are involved in the regulation of iron and ROS metabolism. Some miRNAs, for example, miRNA-675 and miRNA-181, promote ferroptosis by suppressing the transcription of Nrf2, unlike other miRNAs such as miRNA-144 and miRNA-153 that instead stimulate the Nrf2 signal pathway resulting in ferroptotic inhibition [186]. Ferroptosis can also modify miRNA levels in various pathologies; specifically, iron accumulation reduces and increases the expression of miRNA-758-3p and miRNA-3074-5p, respectively, with the consequent augmentation of osteoblastic apoptosis [187,188]. Oxidative stress due to iron overload inhibits osteoblastic functions and induces osteoporosis by downregulating miRNA-455-3p in osteoblasts. This miRNA participates in osteoporosis through one of its targets, histone deacetylases 2 (HDAC2), a pivotal mediator of osteoblastic differentiation that is overexpressed in the presence of low levels of miRNA-455-3p. In MC3T3-E1 cells treated with ferric ammonium citrate, high levels of HDAC2 due to the decreased expression of miRNA-455-3p inhibit the acetylation of Nrf2 and subsequent activation of Nrf2/antioxidant response elements signaling, which plays a protective role against oxidative stress. All this increases oxidative stress, and reduces osteoblastic growth and differentiation, exacerbating osteoporotic processes [189].

Oxidative stress, in addition to inhibiting osteoblastic growth, upregulates the expression of miRNA-138, involved in the apoptosis of osteoblasts in osteoporosis, by downregulating the inhibitor of metalloproteinase TIMP-1 [190]. However, other studies performed in H_2_O_2_-treated MC3T3-E1 cells highlighted a decrease in miRNA-708 and miRNA-214, and an increase in apoptotic processes. In addition, the upregulation of PTEN and activating transcription factor 4 (ATF4), targets of miRNA-708 and miRNA-214, respectively, occur [191,192]. Therefore, miRNA-708 and miRNA-241 can protect osteoblasts against damage due to oxidative stress by regulating PTEN and AFT4, multifunctional molecules involved in various cellular processes [180]. Table 1 summarizes the targets and effects of the principal miRNAs dysregulated by oxidative stress and inflammation in osteoporosis.

## 6. Conclusions

Under normal conditions, bones undergo continuous remodeling due to bone formation and bone resorption. In this process, osteoblasts and osteoclasts play finely balanced roles supported by biochemical signaling pathways. Hormones, growth factors, ROS, and cytokines induce biological responses in osteoblasts and osteoclasts that are responsible for bone resorption and formation. miRNAs also affect bone formation by regulating the proliferation and differentiation of cells involved in bone homeostasis. Osteoporosis is a silent bone disease characterized by a loss of bone mass and mineral density. The oxidative stress, inflammation, and dysregulation of miRNA expression can contribute to the development of postmenopausal osteoporosis due to the deficiency of estrogen. ROS and proinflammatory cytokines, by controlling transcriptional factors, can down- or upregulate miRNA expression, causing an imbalance in osteoblastogenesis and/or osteoclastogenesis with a consequent alteration of bone formation. However, the deregulated expression of miRNAs by stimulating or inhibiting signaling pathways can also induce oxidative stress and inflammation, resulting in an increase in bone resorption. The interplay among oxidative stress, inflammation, and miRNA expression can be advantageous for identifying new suitable targets to address potential therapeutic strategies. In fact, it could be useful to project molecules that, in addition to having antioxidant and anti-inflammatory proprieties, can improve and/or restore the balance of bone remodeling by normalizing both redox-regulated pathway signaling and the levels of miRNA expression. Compounds able to re-establish normal levels of dysregulated miRNAs in osteoporosis can also offer therapeutic benefits by reducing oxidative and inflammatory damage, and alleviating osteoporotic processes by restoring bone homeostasis.

## Figures and Tables

**Figure 1 ijms-24-03772-f001:**
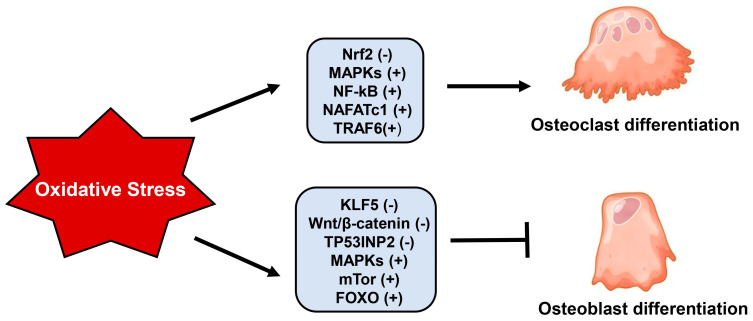
Principal molecular factors involved in the role of oxidative stress on osteoclastic and osteoblastic differentiation. Abbreviations: Nrf2, nuclear factor erythroid-2 related factor; MAPKs, mitogen-activated protein kinases; NF-kB, nuclear factor kappa B; NAFATc1, nuclear factor of activated T-cell cytoplasmic 1; TRAF-6, TNF receptor associated factor 6; KLF5, Krüppel-like factor 5; TP53INP2, tumor protein p53-inducible nuclear protein 2; mTor, mammalian target of rapamycin; FOXO, forkead proteins; (−) downregulated; (+) upregulated.

**Table 1 ijms-24-03772-t001:** Principal miRNAs related to oxidative stress and inflammation in osteoporosis.

miRNA	miRNA Target	Effect	Species	Ref.
miRNA-705 (+)	FOXO1 (−)	Oxidative damage	Osteoporotic mouse BMSCs	Liao et al. [172]
miRNA-182 (+)	FOXO3 (−) Maml1 (−)	TNFα-induced osteoclastogenesis	Murine BMSCs	Miller et al. [173]
miRNA-128 (+)	Sirt-1 (−) NF-kB (+)	Osteoclasogenesis	Mouse bone-marrow-derived macrophages	Shen et al. [174]
miRNA-128-3p (+)	Sirt-1 (−)	Exacerbation of TNFα inflammation	Human BMSCs	Wu et al. [175]
miRNA-141 (+)	ZMPSTE24 (−)	Alteration bone formation	Mice	Bergo et al. [179]
miRNA-320 (+)	(PTEN) (−) PI3K/AKT (−) Osteoblastic genes and genes involved in redox homeostasis (−)	Osteoclastogenesis Oxidative damage Oxidative stress and reduced osteoblastic differentiation and functionality	RAW 264.7 cells MC3T3-E1 cells Human primary osteoblasts	Chen et al. [180] Kong et al. [182] De-Ugarte et al. [183]
miRNA-138 (+)	TIMP-1 (−)	Apoptosis osteoblasts	MC3T3-E1 cells	Yan et al. [190]
miRNA-455-3p (−)	HDAC2 (+)	Inhibition of Nfr2 Oxidative stress	MC3T3-E1 cells	Zhang et al. [189]
miRNA-708 (−)	PTEN (+)	Apoptosis osteoblast Oxidative stress	MC3T3-E1 cells	Zhang et al. [191]
miRNA-214 (−)	(ATF4) (+)	Apoptosis osteoblast Oxidative stress	MC3T3-E1 cells	Lu et al. [192]

Abbreviations: FOXO, forkead proteins; NF-kB, nuclear factor kappa B; Maml1, mastermind-like protein 1, Sirt-1, sirtuin-1; ZMPSTE24, zinc metalloproteinases; PTEN, phosphatase and tensin homolog; TIMP1, metalloproteinase inhibitor 1; HDAC2, histone deacetylase 2; ATF4, activating transcription factor 4. (−) downregulated; (+) upregulated.

## Data Availability

Not applicable.

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
