# Peer review of "Oxidative Stress and Inflammation in Osteoporosis: Molecular Mechanisms Involved and the Relationship with microRNAs"

_ijms, 2023, doi:10.3390/ijms24043772_

Round 1
Reviewer 1 Report
The authors have written a comprehensive review describing the role of oxidative stress and miRNAs in osteoporosis.
I have one suggestion.
At line 420, the authors describe the role of iron-overload in osteoporosis. Here, the link to ferroptosis has not been made, however, would be interesting to include in the manuscript. Several recent studies suggest ferroptosis as a new therapeutic target in osteoporosis. Also, a link between miRNA targeting ferroptotic genes has been made.
Author Response
The authors have written a comprehensive review describing the role of oxidative stress and miRNAs in osteoporosis.
I have one suggestion.
At line 420, the authors describe the role of iron-overload in osteoporosis. Here, the link to ferroptosis has not been made, however, would be interesting to include in the manuscript. Several recent studies suggest ferroptosis as a new therapeutic target in osteoporosis. Also, a link between miRNA targeting ferroptotic genes has been made.
Answer: Ferroptosis was added in the text where indicated by the reviewer. The relative references were added [184-188].

Reviewer 2 Report
The review by Iantomasi et al. emphasizes the mechanisms through which oxidative stress and inflammation affect bone mass, and the relationship between inflammation, oxidative stress, and miRNAs in the pathogenesis of osteoporosis. Both, the topic and the main idea of the manuscript are up-to-date and bring a clear insight into the issue. Even if clear, detailed, and logically organized, the manuscript has some minor points to be fixed:
1. Typos regarding miRNA nomenclature should be checked and corrected, e.g. MiRNA (line 358); hyphens in miRNA-133, 241 (lines 363, 436); miRNa (Table 1) etc.
2. You should mention superoxide dismutase as one of the important enzymes involved in ROS elimination (lines 130-135).
3. Since bone changes in diabetes mellitus have different pathophysiological causes compared to osteoporosis, it is more appropriate to name this complication as "diabetic bone disease", see for example PMID: 31523379, PMID: 35672565.
4. The review is also aimed at identifying targets that facilitate the development of new potential therapeutic methods for the treatment of osteoporosis, therefore it would be appropriate to state at the end of chapters 2 and 3 that many natural molecules have antioxidant and anti-inflammatory properties and could represent a potential in the treatment of osteoporosis, see for example PMID: 32991310, PMID: 35276879, PMID: 33895073.
5. There could be a note (maybe at the end of chapter 3) that there is a close connection between immunity and the gut microbiota, and that the gut microbiome affects bone homeostasis and bone-related diseases, see for example PMID: 36290306, PMID: 28965190.
Author Response
The review by Iantomasi et al. emphasizes the mechanisms through which oxidative stress and inflammation affect bone mass, and the relationship between inflammation, oxidative stress, and miRNAs in the pathogenesis of osteoporosis. Both, the topic and the main idea of the manuscript are up-to-date and bring a clear insight into the issue. Even if clear, detailed, and logically organized, the manuscript has some minor points to be fixed:
- Typos regarding miRNA nomenclature should be checked and corrected, e.g. MiRNA (line 358); hyphens in miRNA-133, 241 (lines 363, 436); miRNa (Table 1) etc.
Answer: Typos regarding miRNA nomenclature were corrected
- You should mention superoxide dismutase as one of the important enzymes involved in ROS elimination (lines 130-135).
Answer: Superoxide dismutase was mentioned
- Since bone changes in diabetes mellitus have different pathophysiological causes compared to osteoporosis, it is more appropriate to name this complication as "diabetic bone disease", see for example PMID: 31523379, PMID: 35672565.
Answer: osteoporosis was replaced with diabetic bone disease. The relative reference was added [81].
- The review is also aimed at identifying targets that facilitate the development of new potential therapeutic methods for the treatment of osteoporosis, therefore it would be appropriate to state at the end of chapters 2 and 3 that many natural molecules have antioxidant and anti-inflammatory properties and could represent a potential in the treatment of osteoporosis, see for example PMID: 32991310, PMID: 35276879, PMID: 33895073.
Answer: At the end of chapters 2 and 3 the potential role of natural molecules with antioxidant and anti-inflammatory properties in the treatment of osteoporosis was added. The relative references were added [102, 103, 146].
- There could be a note (maybe at the end of chapter 3) that there is a close connection between, and that the gut microbiome affects bone homeostasis and bone-related diseases, see for example PMID: 36290306, PMID: 28965190.
Answer: At the end of chapter 3 the link between immunity and the gut microbiota, and the role of gut microbiota on bone health were added. The relative references were added [144, 145].
